# DICR: Direct Intra-image Contrastive Regularization for Contrastive Learning

## Abstract

Typical contrastive self-supervised learning methods apply inter-image contrast to post-projector embeddings, thereby indirectly encouraging the pre-projector representations' invariance to several augmentation operators.[1] While effective, these methods do not account for the inherent difference between semantics-altering (such as cropping and cutout[2]) and semantics-preserving augmentation operators (such as resizing, flipping and color distortion), and thereby lack an explicit mechanism to encourage distinguishable representations for semantically different contents within the same image. We explain, both in reason and in practice, that these issues can harm the generalizability of the representations in downstream tasks. To address these issues, we propose **D**irect **I**ntra-image **C**ontrastive **R**egularization (DICR), a plug-and-play regularization method that directly applies intra-image contrast to pre-projector representations. Empirical results show that DICR can significantly enhance the generalizability of existing methods in downstream tasks, and validate the crucial role of semantic content distinguishability in the generalizable performance of contrastive learning.

## 1 Introduction

Recently, contrastive self-supervised learning has emerged as a powerful paradigm for learning generic representations from unlabeled datasets (He et al., 2020; Grill et al., 2020; Chen & He, 2021; Caron et al., 2021; Bardes et al., 2022; Geiping et al., 2023; Zhang et al., 2024; Gui et al., 2024). These approaches primarily focus on inter-image contrast, which aims to attract views of the same image while repulsing views from different images (Wu et al., 2018; Oord et al., 2018; Hjelm et al., 2018; Bachman et al., 2019; Tian et al., 2020; Yeh et al., 2022). Contrastive learning have demonstrated remarkable results in downstream tasks by indirectly encouraging the representations' invariance to a handful of different augmentation operators indiscriminately (Chen et al., 2020).

However, these augmentation operators are born different. Loosely speaking, some augmentation operators such as resizing, flipping, and color distortion do not alter the semantics of an input image, and thus it is desirable to enforce their invariance in (semantic) representations; other operators such as cropping and cutout do change semantic content, and thereby a mechanism is expected to prevent them from collapsing into a single identical representation. To better understand this intuitive idea, we first explicitly explain it in reason (Section 2.1), and then motivate it from empirical observation perspective (Section 2.2).

Dealing with fundamentally different things indiscriminately can lead to serious problems. In the context of contrastive learning, it will confuse correlation with identity between augmented views of an input image. Concretely, this problem may occur in two cases: when one view is from foreground object and the other is from background (as illustrated in Figure 1a), or when two views show different parts of the foreground object (as illustrated in Figure 1b). This confusion in representation learning will probably lead to performance degeneration on downstream tasks where the correlation does not necessarily hold (for example in Figure 2).

---

[1]By default, we refer 'representations' to the pre-projector representations used in downstream tasks, while we define 'embeddings' as the post-projector representations that are used during pretraining.

[2]We focus on cropping in the experiments because it is a more commonly used and crucial augmentation operator in contrastive learning. Its significance has been empirically demonstrated by Chen et al. (2020) and theoretically explained by Wang et al. (2021).

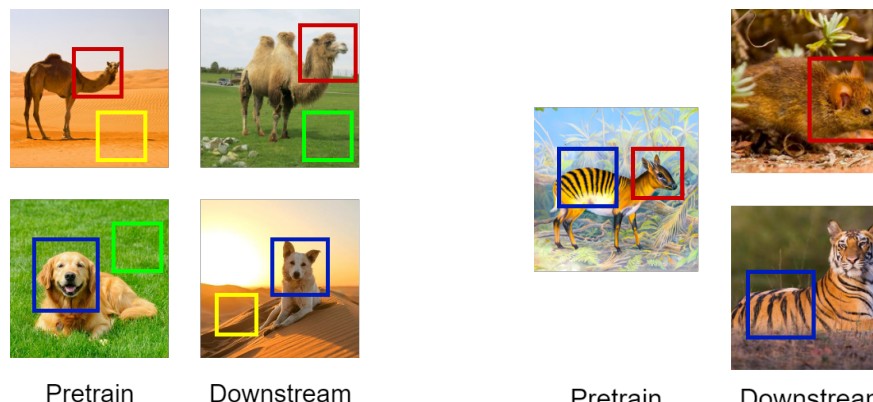

(a) Attracting foreground and background into identical representations can confuse downstream models.

(b) Attracting different parts of the same object into the same representations can mislead downstream models.

Figure 1: Illustration of two kinds of correlation and their effects on the generalization of contrastive learning. The boxes in the same color represent views with similar semantics.

To tackle this problem, we develop a regularization method, called **D**irect **I**ntra-image **C**ontrastive **R**egularization (DICR, pronounced duiker[3]). It explicitly tries to guarantee the distinguishability of different but co-occurring contents via intra-image contrast on representations, and also preserve their correlation via inter-image contrast on embeddings. Our contribution is threefold:

- Firstly, we identify that treating semantics-altering and semantics-preserving augmentation operators indiscriminately can be suboptimal for downstream tasks, as it confuses correlation with identity, and does not prevent representations of different yet co-occurring contents from collapsing into a single identical representation.

- Secondly, as existing methods lack a mechanism to address this issue, we propose DICR, which adds no extra parameters and can efficiently regularizes existing methods.

- Finally, our empirical results demonstrate DICR's capacity to differentiate between different semantic contents within an image, and its effectiveness in improving the generalizable downstream performance of existing methods.

## 2 MOTIVATION

### 2.1 EXPLICIT EXPLANATION IN REASON

Typical inter-image contrastive learning methods attract the post-projector embeddings between views from the same image to model the invariance to certain augmentation operators. Such direct attraction on post-projector embeddings can lead to an indirect attraction on the pre-projector representations, thus encouraging, though not guaranteeing, invariance of the pre-projector representations to these augmentation operators.

However, augmentation operators are inherently different, and encouraging invariance indiscriminately to all the operators can be suboptimal. On the one hand, some augmentation operators are semantics-preserving, such as resizing, flipping, and color distortion. Modeling invariance to these operators is tantamount to modeling semantic identity, which can help downstream models focus on core semantic features instead of spurious style features (Mitrovic et al., 2020). On the other hand, other augmentation operators, such as cropping and cutout, are semantics-altering. Modeling invariance to them involves modeling the correlation between different yet co-occurring semantic contents through attracting their representations. Although the correlation between co-occurring contents does exist in the real world, the co-occurrence does not always hold true under all circumstances. For instance, camels typically roam in the sandy deserts, and dogs usually frolic in the green grass.

---

[3]A duiker is a small to medium-sized antelope as depicted in Figure 1b.

However, as illustrated in Figure 1a, when camels appear amidst a grass background and dogs appear amidst a desert background in the downstream task, attracting the foreground and the background to an identical representation during pretraining can lead to confusion in downstream models. Another example to consider involves the correlation between object parts (Figure 1b). During pretraining, many images of zebra duikers are included, while in the downstream tasks, images of mice and tigers are included. Similar features exist between the head of a mouse and the head of a duiker, as well as between the back of a tiger and the back of a duiker. Therefore, indistinguishable pretrained representations of the duiker's head and its striped back can mislead downstream models when classifying mice and tigers.

Based on the above explicit analysis, in order to obtain more generalizable representations, it is crucial to develop an explicit mechanism to decouple identity and correlation, thereby ensuring distinguishable representations for different semantic contents.

## 2.2 IMPLICIT EMPIRICAL DEMONSTRATION

Our insight stems from two observations. The first observation is that lower layers have more distinguishable (less invariant) representations for augmentation operators, especially for cropping (Figure 2a, solid lines). The second observation is that some lower layers have more generalizable representations than higher layers (Figure 2b, solid lines). To measure the generalizability, we pretrain ResNet-18 through SimCLR on STL10 and report in-distribution (STL10) and out-of-distribution (CIFAR100) linear readout of each residual block's representations and the post-projector embeddings.

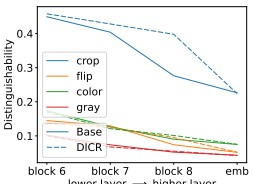

(a) Distinguishability to each augmentation.

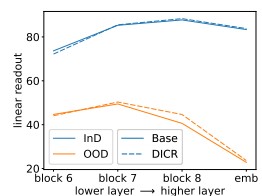

(b) InD and OOD downstream performance.

Figure 2: Distinguishability and downstream performance of different layers.

As for the distinguishability to each augmentation operator separately, we first apply one-vs.-rest augmentation to images in the pretraining dataset to generate views. As depicted in Figure 6, for each image $x$, we generate 10 views through keeping other augmentations constant and applying a specific augmentation operator $\text{aug}(\cdot)$ 10 times to the image. Then, for all the views generated by the given one-vs.-rest augmentation $\left\{v_{\text{aug}(x)}\right\}_x$, we measure the distinguishability of each layer $l$ by: $\text{NormStd}\left(\left\{\boldsymbol{h}^l_{v^{\text{aug}(x)}}\right\}_x\right) = \mathbb{E}_x\left[\sigma_{v^{\text{aug}(x)}}\left[\boldsymbol{h}^l_{v^{\text{aug}(x)}}\right]/\sigma_v\left[\boldsymbol{h}^l_v\right]\right]$, where the std in the numerator $\sigma_x\left[\boldsymbol{h}^l_{v^{\text{aug}(x)}}\right]$ is calculated as the expectation of L2 distance between each representation and the mean representation across the 10 views from the same image. And to make the std comparable across different layers, it is normalized by the std in the denominator that is computed across all views generated by all augmentation operators on the entire dataset. The NormStd can depict the degree of representation dispersion at each layer for a specific augmentation operator. The larger the NormStd is, the more dispersed the representations are for views generated by the augmentation operator, thereby making the representations more distinguishable for the augmentation operator.

As observed in Figure 2a (solid lines), lower layers are better at differentiating different contents compared with higher layers, as they exhibit a larger difference between the NormStd of cropping and the NormStd of semantics-preserving augmentation operators. As observed in Figure 2b (solid lines), some lower layers perform better than higher layers. Pre-projector representations (residual block 8) outperform the post-projector embeddings on both InD (in-distribution) and OOD (out-of-distribution) datasets. Additionally, although residual block 7 is not the final residual block, it surpasses residual block 8 on the OOD dataset.

We empirically observe an association between the representations' capacity to differentiate different semantic contents and their generalizability in both InD (train-test shifts) and OOD (pretrain-train-test shifts) settings, which echoes our explicit explanation in Section 2.1. However, such an effect in existing methods is implicit, so we aim to make it explicit by introducing DICR, which decouples identity and correlation by using representations to model identity and embeddings to model correlation,

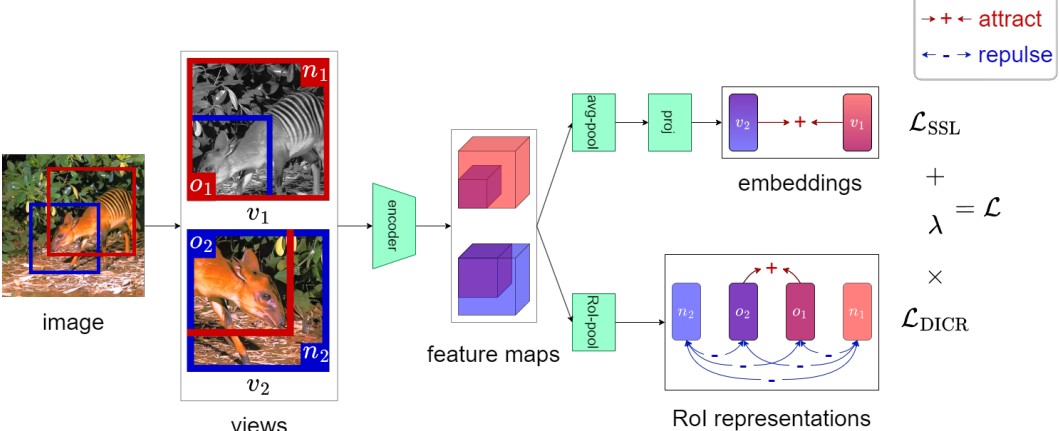

Figure 3: The overall framework of DICR consists of two branches. In the top branch, typical inter-image contrastive learning such as SimCLR (Chen et al., 2020) or SimSiam (Chen & He, 2021) is performed on embeddings. In the bottom branch, intra-image contrast is directly applied to RoI representations. DICR decouples the modeling of identity and correlation, and thereby explicitly promotes different representations for different semantic contents within each image.

respectively. We also include the results of DICR as dashed lines in Figure 2. DICR significantly promotes distinguishable representations of different semantic contents within an image, and achieves more generalizable performance than the baseline on both residual block 8 and residual block 7.

## 3 METHOD

The overall framework of **D**irect **I**ntra-image **C**ontrastive **R**egularization (DICR) is illustrated in Figure 3. The framework consists of two branches, where the first branch is the typical inter-image contrastive learning method, and the second branch is our proposed DICR. We start by reviewing how inter-image contrastive learning methods work and why they confuse identity with correlation (Section 3.1). Next, we introduce DICR and describe how it explicitly addresses the issues (Section 3.2).

### 3.1 INTER-IMAGE CONTRASTIVE SELF-SUPERVISED LEARNING

The typical view generation process begins with sampling an image, followed by applying augmentation operators such as random resized cropping, flipping, and color distortion to obtain a positive view pair $v_1, v_2$. From the view pair, we identify four regions: $o_1, o_2, n_1, n_2$. Here, the subscript $i$ indicates that the region is from the view $v_i$. $o_1, o_2$ represents the overlapping region between $v_1, v_2$, while $n_1, n_2$ represents the non-overlapping region. As illustrated in Figure 3, the overlapping regions $o_1, o_2$ always contain the same semantic content but with potentially different styles, while region pairs other than $o_1, o_2$ can contain semantically different contents. Then, the view pair $v_1, v_2$ is processed by a shared encoder (e.g., ResNet (He et al., 2016)) to obtain feature maps $\boldsymbol{f}_1, \boldsymbol{f}_2$ with spatial and channel dimensions.

Inter-image contrastive learning methods (Chen et al., 2020; He et al., 2020; Chen & He, 2021; Zhang et al., 2024) apply global average pooling to $\boldsymbol{f}_1, \boldsymbol{f}_2$ to obtain the representation $\boldsymbol{h}_i$. Here, the global average pooling operation can be seen as approximating the expectation of the entire feature map through sampling:

$$\boldsymbol{h}_i = \mathbb{E}_{v_i}[\boldsymbol{f}_i] = \frac{\int_{v_i} \boldsymbol{f}_i \cdot dS}{\int_{v_i} dS} \approx \mathrm{AvgPool}(\boldsymbol{f}_i), \ i \in \{1, 2\}, \tag{1}$$

where the representation $\boldsymbol{h}_i$ captures the average features of the entire view $v_i$, as the expectation is taken over the whole view. Then these representations are fed into a projection head to obtain

embeddings $\boldsymbol{z}_1, \boldsymbol{z}_2$, and a loss function, such as InfoNCE loss (Oord et al., 2018), is minimized to attract the positive view pair to similar embeddings:

$$\mathcal{L}_{\text{InfoNCE}} = -t \log \frac{\exp(\cos(\boldsymbol{z}_1, \boldsymbol{z}_2)/t)}{\exp(\cos(\boldsymbol{z}_1, \boldsymbol{z}_2)/t) + \sum_{i=1}^{N} \exp(\cos(\boldsymbol{z}_1, \boldsymbol{z}_i^-)/t)}, \tag{2}$$

where $\cos(\cdot)$ denotes the cosine similarity, $\{\boldsymbol{z}_i^-\}_{i=1}^{N}$ denotes $N$ randomly sampled negative embeddings, and $t$ is a temperature parameter.

Optimizing this loss function can lead to confusion between identity and correlation. We will use a simplified model to explain this. In fact, based on the formulation in Equation 1, the representation $\boldsymbol{h}_i$ can be decoupled into representations that capture features of specific regions $o_i, n_i$:

$$\boldsymbol{h}_i = \frac{\int_{v_i} \boldsymbol{f}_i \cdot dS}{\int_{v_i} dS} = \frac{\int_{o_i+n_i} \boldsymbol{f}_i \cdot dS}{\int_{v_i} dS} = \frac{S_{o_i} \boldsymbol{h}_{o_i} + S_{n_i} \boldsymbol{h}_{n_i}}{S_{v_i}}, \; i \in \{1, 2\}, \tag{3}$$

where $S_{o_i}, S_{n_i}, S_{v_i}$ denote the areas of regions $o_i, n_i, v_i$, and $\boldsymbol{h}_{o_i}, \boldsymbol{h}_{n_i}$ denote the representations of regions $o_i, n_i$. Assuming a linear projection head $\boldsymbol{z} = \boldsymbol{W}\boldsymbol{h}$, based on the representation decoupling in Equation 3, the similarity of the embeddings can be further decoupled into:

$$\cos(\boldsymbol{z}_1, \boldsymbol{z}_2) = \frac{\overbrace{S_{o_1} S_{o_2} \cdot \boldsymbol{h}_{o_1}^\top \boldsymbol{W}^\top \boldsymbol{W} \boldsymbol{h}_{o_2}}^{\text{Identity term.}} + \overbrace{\sum_{\{x,y\} \in \mathcal{N}} S_x S_y \cdot \boldsymbol{h}_x^\top \boldsymbol{W}^\top \boldsymbol{W} \boldsymbol{h}_y}^{\text{Correlation term.}}}{\|\boldsymbol{z}_1\| \|\boldsymbol{z}_2\| S_{v_1} S_{v_2}}, \tag{4}$$

where $\mathcal{N} = \{\{o_1, n_2\}, \{n_1, o_2\}, \{n_1, n_2\}\}$ denotes the non-intersecting region pairs between two views. The identity term in Equation 4 encourages the use of representations to capture the identity of identical contents with different styles. Take the duiker image in Figure 3 as an example. Both overlapping regions $o_1, o_2$ cover the duiker's head, but $o_1$ is smaller and grayscale, whereas $o_2$ is larger and flipped. To maximize the identity term, the embeddings of $o_1, o_2$ should remain invariant under these semantic-preserving augmentations, thereby encouraging the representations $\boldsymbol{h}_{o_1}, \boldsymbol{h}_{o_2}$ to capture identical semantic contents. The correlation term in Equation 4 encourages the use of representations to model the correlation between different image contents. For instance, in Figure 3, the non-overlapping region $n_1$ covers the duiker's body, while the non-overlapping region $n_2$ primarily covers the background. Therefore, maximizing the correlation term encourages similar representations for the correlated semantic contents, i.e., the duiker's head, the body, and the background.

Inter-image contrastive learning confuses identity with correlation by using the same mechanism to model both, and there is no explicit mechanism to prevent the representations of different contents from collapsing into a single identical representation. However, such distinguishability is necessary for representations to be generalizable, as discussed in Section 2.

## 3.2 DIRECT INTRA-IMAGE CONTRASTIVE REGULARIZATION

We propose DICR to decouple identity and correlation. The basic idea behind DICR is that the similarity in representation space should reflect the identity between contents, while the similarity in embedding space should reflect the correlation between contents. Therefore, besides the typical inter-image contrastive loss on embeddings, there should be a mechanism to ensure different yet co-occurring contents have different representations. A straightforward implementation of this idea would be to sample some semantically identical views as positive views and different yet co-occurring views as negative views, then apply a contrastive loss to their representations as a form of regularization. However, this implementation can be computationally slow, as it does additional forward and backward propagation to optimize the regularization term. In fact, as discussed in Section 3.1, there are ready-made regions with semantically identical and different yet co-occurring contents in views generated by the typical view generation process, and their representations can

be derived by decoupling representations as described in Equation 3. Therefore, DICR can be implemented as:

$$\mathcal{L}_{\text{DICR}} = -\tau \log \frac{\exp(\text{sim}(\boldsymbol{h}_{o_1}, \boldsymbol{h}_{o_2})/\tau)}{\exp(\text{sim}(\boldsymbol{h}_{o_1}, \boldsymbol{h}_{o_2})/\tau) + \sum_{\{x,y\} \neq \{o_1, o_2\}} \exp(\text{sim}(\boldsymbol{h}_x, \boldsymbol{h}_y)/\tau)}, \tag{5}$$

where $\text{sim}(\boldsymbol{h}_{o_1}, \boldsymbol{h}_{o_2})$ denotes the similarity between overlapping regions' representations $\boldsymbol{h}_{o_1}, \boldsymbol{h}_{o_2}$, while $\{\text{sim}(\boldsymbol{h}_x, \boldsymbol{h}_y)\}_{\{x,y\} \neq \{o_1, o_2\}}$ denotes the similarity of representations between region pairs other than $\{o_1, o_2\}$, and the temperature $\tau$ is a hyperparameter controlling the softness of $\mathcal{L}_{\text{DICR}}$. For the overlapping region $o_i$, which is always rectangular, we can directly apply RoIAlign to the feature map $\boldsymbol{f}_i$ to approximate the representation $\boldsymbol{h}_{o_i}$:

$$\boldsymbol{h}_{o_i} = \mathbb{E}_{o_i}[\boldsymbol{f}_i] = \frac{\int_{o_i} \boldsymbol{f}_i \cdot dS}{\int_{o_i} dS} \approx \text{RoIAlign}_{o_i}(\boldsymbol{f}_i). \tag{6}$$

As for the non-overlapping region $n_i$, which is not necessarily rectangular, we can acquire its representation $\boldsymbol{h}_{n_i}$ by substituting the approximations of $\boldsymbol{h}_{o_i}, \boldsymbol{h}_i$ into Equation 3. We use cosine similarity to measure the similarity between two representations. However, there are three edge cases to consider when the region areas $S_{o_i}, S_{o_j}$ are trivial:

- For completely non-overlapping views $v_1, v_2$, we should repulse their representations.

- For completely overlapping views $v_1, v_2$, we should attract their representations.

- When one view $v_i$ is strictly contained in the other view $v_j$, we should attract the overlapping region's representation and repulse the pairs $\{\boldsymbol{h}_{o_i}, \boldsymbol{h}_{n_j}\}$ and $\{\boldsymbol{h}_{o_j}, \boldsymbol{h}_{n_j}\}$.

To handle these edge cases we introduce a hyperparameter $\epsilon$. If both region areas are less than $\epsilon$, we set the similarity to 1. Conversely, if only one area is less than $\epsilon$ and the other area is greater than or equal to $\epsilon$, we set the similarity to 0.[4] The final similarity function is defined as:

$$\text{sim}(\boldsymbol{h}_x, \boldsymbol{h}_y) = \begin{cases} \cos(\boldsymbol{h}_x, \boldsymbol{h}_y), & \text{if both } S_x \text{ and } S_y \geq \epsilon \\ 1, & \text{if both } S_x \text{ and } S_y < \epsilon \\ 0, & \text{if either } S_x \text{ or } S_y \text{ but not both } < \epsilon \end{cases}. \tag{7}$$

The overall objective is formulated as the weighted sum of the inter-image contrastive loss $\mathcal{L}_{\text{SSL}}$ and our proposed regularization term $\mathcal{L}_{\text{DICR}}$:

$$\mathcal{L} = \mathcal{L}_{\text{SSL}} + \lambda \mathcal{L}_{\text{DICR}}, \tag{8}$$

where $\lambda$ is a hyperparameter controlling the weight of $\mathcal{L}_{\text{DICR}}$. We adopt a simple warm-up strategy, which initializes $\lambda$ to 0 and increases $\lambda$ linearly every epoch, to avoid occasional training failures in the early stages of training.

## 4 EXPERIMENTS

In this section, we empirically evaluate the effectiveness of DICR. We first demonstrate the superior generalizability of DICR by comparing it with existing contrastive learning methods. Then, we investigate the behavior of DICR through analytical experiments.

---

[4]We assume that the representations of null regions have the same direction, and assume that the representations between a null region and a meaningful region are orthogonal.

Table 1: Linear readout accuracy (%) on the in-distribution datasets and out-of-distribution datasets. The best results are highlighted in bold.

| Pretrain | Evaluate | SimCLR | | MoCo[6] | | SimSiam | | Matrix-SSL | |
|---|---|---|---|---|---|---|---|---|---|
| | | Base | DICR | Base | DICR | Base | DICR | Base | DICR |
| CIFAR10 | CIFAR10 | 89.130 | **90.150** | 90.410 | **91.210** | **90.580** | **90.580** | 91.430 | **92.280** |
| | CIFAR100 | 49.090 | **53.200** | 47.550 | **52.690** | 48.550 | **50.560** | 47.370 | **53.670** |
| | Tiny200 | 28.920 | **31.850** | 26.490 | **30.880** | 28.080 | **28.700** | 26.010 | **31.670** |
| | STL10 | 75.338 | **76.600** | 76.412 | **78.100** | 75.600 | **75.862** | 76.287 | **77.075** |
| CIFAR100 | CIFAR100 | 60.860 | **62.570** | 62.190 | **63.300** | 63.390 | **64.220** | 66.590 | **66.680** |
| | CIFAR10 | 76.170 | **79.260** | 75.320 | **77.920** | 77.530 | **79.150** | 78.390 | **80.660** |
| | Tiny200 | 31.490 | **34.440** | 30.590 | **33.840** | 31.800 | **33.130** | 31.930 | **35.260** |
| | STL10 | 66.525 | **67.838** | 66.650 | **67.912** | 66.812 | **67.700** | 66.213 | **69.175** |
| Tiny200 | Tiny200 | 44.030 | **44.990** | **48.280** | 47.680 | 41.990 | **42.150** | 45.280 | **45.550** |
| | CIFAR10 | 71.620 | **72.100** | 72.160 | **74.430** | 68.410 | **69.950** | 69.810 | **72.520** |
| | CIFAR100 | 46.830 | **48.740** | 47.740 | **50.610** | 36.650 | **39.410** | 39.140 | **43.290** |
| | STL10 | 73.112 | **75.550** | 75.888 | **76.400** | 70.650 | **71.537** | 72.338 | **73.088** |
| STL10 | STL10 | 87.750 | **88.338** | 89.188 | **89.475** | 86.737 | **87.088** | 88.312 | **88.862** |
| | CIFAR10 | 72.210 | **74.700** | 74.030 | **75.490** | 65.130 | **69.090** | 68.140 | **72.530** |
| | CIFAR100 | 40.590 | **44.630** | 43.200 | **45.300** | 23.540 | **27.420** | 24.350 | **34.610** |
| | TinyImagenet | 37.940 | **40.700** | 39.720 | **40.970** | 28.490 | **29.950** | 28.260 | **32.950** |

## 4.1 DICR ENHANCES GENERALIZABILITY IN DOWNSTREAM TASK PERFORMANCE.

**Pretraining.** We conduct experiments on CIFAR10/CIFAR100 (Krizhevsky, 2009), TinyImagenet (Le & Yang, 2015), and STL10 (Coates et al., 2011). We consider SimCLR (Chen et al., 2020), MoCo (He et al., 2020), SimSiam (Chen & He, 2021) and MatrixSSL (Zhang et al., 2024) as baselines. Following Chen et al. (2020), we set the augmentation strategy to resized cropping, flipping, and color distortion. We adopt ResNet-18 (He et al., 2016) as the backbone for all the experiments, modifying it by removing the first max pooling operation and replacing the first 7x7 convolutional layer of stride 2 with a 3x3 convolutional layer of stride 1, to accommodate the smaller image sizes in our selected datasets compared to ImageNet. For optimization, we use the SGD optimizer with momentum 0.9 and weight decay $1 \times 10^{-4}$, and perform cosine-annealing learning rate scheduling. For CIFAR10 and CIFAR100, we initialize the learning rate as 0.5 and pretrain the models for 500 epochs with batch size 512. For TinyImagenet and STL10, we initialize the learning rate as 0.25 and pretrain the models for 250 epochs with batch size 256.

Regarding our approach, we adjust the hyperparameters for each baseline individually, due to their distinct loss functions. For SimCLR, we assign a final weight $\lambda$ of 40 and a temperature $\tau$ of 0.05 for all datasets. In the case of MoCo, we set a final weight $\lambda$ of 10 and a temperature $\tau$ of 0.02 across all datasets. For SimSiam, we determine a final weight $\lambda$ of 2 and a temperature $\tau$ of 0.005 for all datasets. In the case of Matrix-SSL, we set a final weight $\lambda$ of 10 and a temperature $\tau$ of 0.02 across all datasets. We configure the threshold $\epsilon$ to $2 \times 2$ pixels for CIFAR10 and CIFAR100, and to $3 \times 3$ pixels for TinyImagenet and STL10.[5]

**Evaluation protocol.** We follow the typical linear readout protocol (He et al., 2020), training a linear classifier on top of the frozen backbone for 100 epochs using the SGD optimizer. We evaluate the representations on in-distribution and out-of-distribution datasets. For each dataset, we evaluate the in-distribution on the pretraining dataset itself, and the out-of-distribution on the other three datasets.

**Main results.** The results are shown in Table 1. In most settings, DICR significantly improves the linear readout accuracy compared to the baselines. The improvements are more pronounced on out-of-distribution datasets, demonstrating DICR's generalizable performance in out-of-distribution downstream tasks.

---

[5]Our code is built upon the implementation of Peng et al. (2022). All the experiments can be run on 2 NVIDIA 3090 GPUs.

## 4.2 ANALYTICAL STUDY

**DICR encourages distinguishable representations for different contents.** The motivation of DICR is to decouple identity and correlation, promoting distinguishable representations for semantically different contents within the same image. In this experiment, we further explore whether the representations achieve this. We use intersection over union (IoU) between positive views as a measure of the amount of the identical contents between views. We then investigate the distinguishability of the representations for view pairs with low IoU. The representations are pretrained on STL10 using SimCLR with and without DICR. The view pairs with different

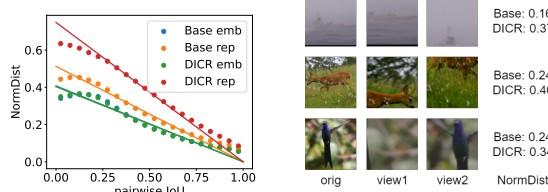

(a) The relationship between IoU and NormDist.

(b) Illustrations of almost non-overlapping view pairs.

Figure 4: Quantitative and qualitative evaluation of content distinguishability.

IoU are generated through one-vs.-rest augmentation (see Figure 6). We derive the formula to measure the pairwise distance by decoupling the NormStd: $\text{NormDist}\left(\boldsymbol{h}^l_{v_i^{\text{crop}(x)}}, \boldsymbol{h}^l_{v_j^{\text{crop}(x)}}\right) = \left\|\boldsymbol{h}^l_{v_i^{\text{crop}(x)}} - \boldsymbol{h}^l_{v_j^{\text{crop}(x)}}\right\| / \sigma_v\left[\boldsymbol{h}^l_v\right]$, where $v_i^{\text{crop}(x)}$ is a specific view generated by cropping the image $x$. NormDist can be interpreted as the distinguishability of layer $l$ for the view pair $v_i^{\text{crop}(x)}, v_j^{\text{crop}(x)}$ with potentially low content overlap (IoU).

Figure 4 shows that DICR increases the representation distance for views with low IoU, while it does not significantly affect the embedding distance compared to the baseline. The results indicate that DICR explicitly promotes distinguishable representations for different contents, while it does not affect the modeling of contents' correlation in the embedding space. We also illustrate some nearly non-overlapping view pairs that exhibit similar representations for the baseline but have significantly different representations for DICR in Figure 4b. The first image shows a foggy ship image. The baseline confuses the ship's aerial and hull due to the fog, while DICR differentiates them. The second image is a deer in grass. The baseline fails to distinguish the deer's neck and legs, but DICR does. The last image is a flying bird. The baseline treats the bird and the bird-less background similarly, but DICR identifies them as different.

**The effect of intra-image contrast in DICR.** The repulsion of non-intersecting representations in DICR is achieved in a contrastive way, as the basic idea behind DICR is that different yet co-occurring contents should have less similar representations than semantically identical contents. In this experiment, we investigate the necessity of contrastive learning in DICR. We propose a variant of DICR called **D**irect **I**ntra-image **R**epulsion **R**egularization (DIRR), where the term $\text{sim}(\boldsymbol{h}_{o_1}, \boldsymbol{h}_{o_2})$ in Equation 5 is replaced with a constant hyperparameter $s$. ResNet-18 models are pretrained using SimCLR

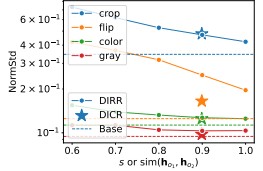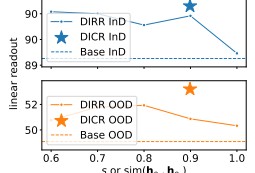

(a) Distinguishability to each augmentation.

(b) InD and OOD downsteram performance.

Figure 5: Distinguishability and downstream performance of DIRR, DICR and baseline.

and DIRR on CIFAR10 dataset. We set $s$ to $0.6, 0.7, 0.8, 0.9, 1$ and set the same other hyperparameters as DICR. Then, the pretrained models are evaluated on in-distribution (CIFAR10) dataset and out-of-distribution (CIFAR100) dataset. As shown in Figure 5b, DIRR performs consistently better than the baseline. We attribute its superiority over the baseline to its direct promotion of

---

[6]We implement the symmetric version of MoCo following Chen et al. (2020) to make $\mathcal{L}_{\text{DICR}}$ more easily optimized.

distinguishable representations for different contents, as shown in Figure 5a. However, it performs consistently worse than DICR. We attribute this to DIRR's inability to model the identity of semantically identical contents with different styles. As shown in Figure 5a, it is significantly less invariant to semantics-preserving augmentation operators (especially for flipping) than DICR and the baseline. These results demonstrate the necessity of contrastive learning in DICR, to correctly model identity.

**The effect of directness in DICR.** In DICR, the regularization is directly applied on the representations, as our goal is to explicitly promote the pre-projector representations' distinguishability to different contents. In this experiment, we study the effect of directness in DICR. We consider a variant of DICR called **I**ntra-image **C**ontrastive **R**egularization (ICR), where the RoI representations are first projected to embeddings by a parameterized projector, then we replace the representations in Equation 5 with these embeddings. We use the same projector architecture as SimCLR, and tune the temperature $\tau$ for DIRR. We set $\tau$ to 0.2, and set other hyperparameters the same as DICR. In Table 2, we observe that DICR outperforms ICR on both in-distribution and out-of-distribution datasets. Additionally, $ICR_2^1$, which uses a linear projector in regularization, achieves the performance closest to that of $DICR_2$. We conjecture that attracting and repulsing within a linear subspace of the representations has a similar effect to directly doing so on the representations.

Table 2: Comparison between DICR and ICR. The superscript of ICR indicates layer count in ICR projectors.

| Method | In-distribution | Out-of-distribuion | | |
|---|---|---|---|---|
| | CIFAR10 | CIFAR100 | TinyImagenet | STL10 |
| Base | 89.130 | 49.090 | 28.920 | 75.338 |
| DICR | **90.150** | **53.200** | **31.850** | **76.600** |
| $ICR^1$ | 90.010 | 52.740 | 31.660 | 76.075 |
| $ICR^2$ | 89.790 | 50.910 | 30.330 | 76.312 |

Table 3: Sensitive analysis of hyperparameters.

(a) Sensitive analysis of $\lambda$.

| $\lambda$ | In-distribution | Out-of-distribution | | |
|---|---|---|---|---|
| | CIFAR10 | CIFAR100 | Tiny200 | STL10 |
| Base | 89.130 | 49.090 | 28.920 | 75.338 |
| 10 | 90.300 | 52.860 | 31.670 | 76.838 |
| 20 | 89.890 | 52.630 | 32.010 | 77.100 |
| 40 | 90.150 | 53.200 | 31.850 | 76.600 |
| 100 | 90.070 | 54.190 | 32.410 | 76.912 |
| 200 | 89.780 | 53.940 | 32.630 | 76.237 |

(b) Sensitive analysis of $\tau$.

| $\tau$ | In-distribution | Out-of-distribution | | |
|---|---|---|---|---|
| | CIFAR10 | CIFAR100 | Tiny200 | STL10 |
| Base | 89.130 | 49.090 | 28.920 | 75.338 |
| 0.01 | 90.080 | 53.110 | 32.170 | 76.938 |
| 0.02 | 90.110 | 52.730 | 31.860 | 76.000 |
| 0.05 | 90.150 | 53.200 | 31.850 | 76.600 |
| 0.1 | 90.260 | 53.170 | 31.130 | 77.700 |
| 0.2 | 89.760 | 48.290 | 28.090 | 76.325 |

**Sensitive analysis of hyperparameters.** We conduct sensitivity analyses of the hyperparameters $\lambda$ and $\tau$ on the CIFAR10 dataset, using SimCLR as the baseline. The results in Table 3 show that DICR can robustly improve downstream performance compared to the baseline.

### 4.3 COMPARISON WITH OTHER IMPLICIT OR EXPLICIT METHODS

In this section, we compare DICR with three other methods that implicitly or explicitly promote distinguishable representations for different contents.

**Comparison with implicit methods.** As observed in Figure 2a, the projector in inter-image contrastive learning can implicitly enhance the distinguishability of representations for different contents. Therefore, we consider employing deeper projectors in SimCLR without other regularization terms as an implicit baseline. The results in Table 4 show that the improvements brought by implicit methods are limited compared to DICR.

**Comparison with explicit methods.** Zhang & Ma (2022) introduces augmentation embeddings to facilitate the projector to explicitly model invariance to a specific augmentation operator (such as cropping), to ensure that useful information is stored in the representations. We compare DICR with this method on CIFAR10 dataset using SimCLR as the baseline, which we refer to as CropEmb. The results in Table 4 show that DICR outperforms CropEmb on both in-distribution and out-of-distribution datasets.

Table 4: Comperison between DICR and other implicit or explicit methods. The subscript of implicit methods indicates the number of layers in SSL projectors.

| Method | | In-distribution | Out-of-distribuion | | |
| --- | --- | --- | --- | --- | --- |
| | | CIFAR10 | CIFAR100 | TinyImagenet | STL10 |
| Base | SimCLR | 89.130 | 49.090 | 28.920 | 75.338 |
| Implicit | SimCLR$_3$ | 89.640 | 51.060 | 30.760 | 75.987 |
| | SimCLR$_4$ | 88.750 | 50.790 | 29.990 | 75.388 |
| Explicit | CropEmb | 89.050 | 49.650 | 31.020 | 75.325 |
| | DICR | **90.150** | **53.200** | **31.850** | **76.600** |

## 5 RELATED WORK

**Contrastive self-supervised learning.** Contrastive learning is a widely adopted self-supervised learning paradigm that aims to learn a generic representation from unlabeled pretraining datasets (He et al., 2020; Grill et al., 2020; Chen & He, 2021; Caron et al., 2021; Bardes et al., 2022; Geiping et al., 2023; Zhang et al., 2024; Gui et al., 2024). The key idea of contrastive learning is to perform inter-image contrast, which is to attract views generated from the same images and to repulse views generated from different images (Wu et al., 2018; Oord et al., 2018; Hjelm et al., 2018; Bachman et al., 2019; Tian et al., 2020; Yeh et al., 2022). Different from inter-image contrast adopted by the above mentioned methods, DICR applies intra-image contrast to existing contrastive learning methods to enhance their generalizability on downstream tasks.

**Learning object-level representations through intra-image contrast.** Some works involve intra-image contrast (Hénaff et al., 2021; Xiao et al., 2021; Wang et al., 2022; Yan et al., 2022) to better align with pixel-wise tasks. The main difference between these methods and DICR is how they contruct positive and negative pairs. Hénaff et al. (2021); Wang et al. (2022) rely on external tools (external segmentation algorithms in Hénaff et al. (2021) and copy-paste in Wang et al. (2022)) to generate positive and negative pairs. The positive view pairs generated by these methods can be different parts of the same object, and thereby do not decouple identity from correlation, which differs from DICR. The work by Yan et al. (2022) adapts contrastive learning for pretraining on anatomical images. It applies global and local pixel-level contrast, involving intra-image pixels as negatives. The correlation between different pixels within single images is somewhat overlooked in Yan et al. (2022), since different pixels are never treated as positive pairs. However, in contrastive learning for regular images, the correlation between different pixels is misleading but can be useful for downstream tasks, so DICR is designed to preserve both the identity and correlation. Xiao et al. (2021) involves both intra-image contrast that attract the same contents with different styles and inter-image contrast that attract different contents from the same image. However, the intra-image contrast in Xiao et al. (2021) is not directly applied on representations, which has been shown to be essential in decoupling identity from correlation in Section 4.2. Additionally, the method by Xiao et al. (2021) handle the edge case where the two views are completely non-overlapping by simply ignoring the repulsion between them, which is different from DICR.

## 6 CONCLUSION

In this work, we identify the importance of decoupling identity and correlation in contrastive learning to enhance the generalizability of downstream performance. We propose DICR, a regularization method that can decouple identity and correlation in existing contrastive learning methods. It apply intra-image contrast on representations, and also preserve their correlation via inter-image contrast on embeddings. Our empirical evidence shows that DICR substantially improves the generalizability of downstream performance in existing methods, underscoring the pivotal role that content distinguishability plays in the robust performance of contrastive learning.

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

## A  ONE-VS.-REST AUGMENTATIONS

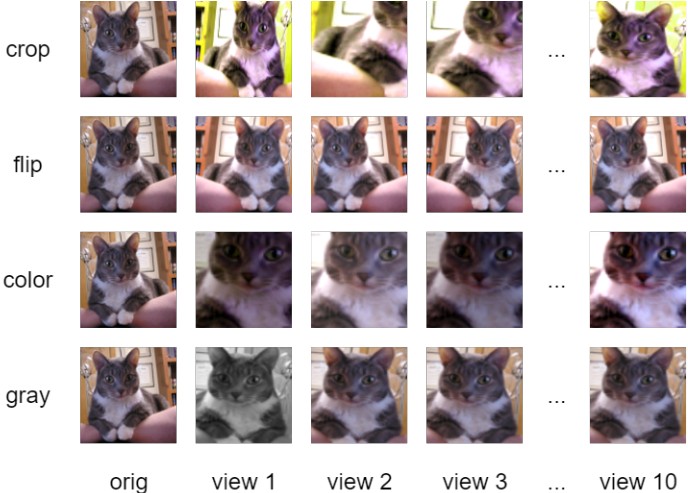

Figure 6: Illustration of one-vs.-rest augmentations. We keep other augmentations constant and apply a selected augmentation aug $(\cdot)$ ten times to the image. This generates ten views $v^{\mathrm{aug}(x)}$, all of which share the same set of other augmentations but vary in the chosen augmentation.

