# OpenReview forum: "DICR: Direct Intra-image Constrastive Regularization for Contrastive Learning"
_ICLR.cc/2025/Conference — ICLR 2025 Conference Withdrawn Submission_

### Official Review · Reviewer_1ESs · 2024-10-31

**Soundness:** 2
**Presentation:** 2
**Contribution:** 2
**Rating:** 5
**Confidence:** 3

**Summary:**

Existing methods typically employ inter-image contrastive learning, indirectly encouraging the representations to be invariant to augmentation operations. However, they fail to differentiate between semantics-altering augmentations (e.g., cropping, cutout) and semantics-preserving augmentations (e.g., resizing, flipping, color distortion). This work proposes a Direct Intra-image Contrastive Regularization (DICR) method to improve the generalizability of contrastive self-supervised learning.

**Strengths:**

1. The motivation of this paper is reasonable and clear.
2. The effectiveness of the proposed regularization was verified on multiple baselines.

**Weaknesses:**

1. Although the motivation is clear, it lacks concrete examples to support the finding that prior contrastive learning methods cause confusion in downstream models. This absence of empirical evidence makes the motivation less solid. Moreover, section 2.2 "Implicit Empirical Demonstration" lacks clarity and is somewhat difficult to understand. Specifically, for example in Figure 2a, it is unclear what "Distinguishability" represents and how this value is derived, as the paper does not provide an explanation for it.
Most importantly, the two observations mentioned in this section—"lower layers have more distinguishable representations" and "lower layers have more generalizable representations"—lack a clear conclusion or insight, making it difficult to understand the connection between these observations and the overall motivation. This part is challenging to follow and comprehend.
2. The proposed Direct Intra-image Contrastive Regularization (DICR)  is somewhat simple and lacks novelty. By simply talking about overlapping parts being close to each other and non-overlapping parts being willing to each other, does this destroy similar feature representations in the non-overlapping parts. As an extreme example given by the authors, if two views of a crop with non-overlapping regions happen to be the left and right ears of a dog, should their representations also be far away from each other in feature space? I think the method is overly simplistic in its design and harder to achieve what the authors are aiming for.

3. Although the performance on image classification tasks was validated, other downstream tasks (such as detection and segmentation) were not tested. It is necessary to further verify the generalizability of DICR on additional downstream tasks.

**Questions:**

1.In Figure 2a, it is unclear what "Distinguishability" represents and how this value is derived.

---

### Official Review · Reviewer_CZZ5 · 2024-11-01

**Soundness:** 3
**Presentation:** 2
**Contribution:** 3
**Rating:** 5
**Confidence:** 3

**Summary:**

The authors propose that  indiscriminately treating semantics-altering and semantics-preserving augmentation operators may be suboptimal for downstream tasks. To address this issue, a intra-image constrastive learning method is designed with no extra parameters, which improves the generalizable downstream performance of existing methods.

**Strengths:**

1. The authors conduct extensive experiments and analytical study, and perform comparisons with explicit and implicit models.
2. The authors provide implicit empirical demonstration.

**Weaknesses:**

1. The motivation for the proposed intra-image contrastive loss is not clear. Why do we attract foreground and background into identical representations? And why do we attract different parts of the same object into the same representations? The Figure.1 and Section 2.1 confuse me.

**Questions:**

1. The authors should carefully explain the motivation of their intra-image contrastive loss.
2. It can be observed in Figure3 that the intra-image contrastive learning divides the image to finer regions and perform attraction and repulsion among n1, n2, o1 and o2. So why does the method still need inter-image constrastive learning?
3. In the results presented in Table 1, why does Base sometimes show better results, and why does Base sometimes show the results same as DICR?

---

### Official Review · Reviewer_NXUz · 2024-11-03

**Soundness:** 3
**Presentation:** 3
**Contribution:** 3
**Rating:** 6
**Confidence:** 4

**Summary:**

The paperintroduces DICR, a new regularization method that improves upon traditional contrastive learning by directly contrasting pre-projector representations within images. DICR enhances the ability of models to generalize to new tasks by distinguishing between semantically different image contents, leading to better performance in downstream applications.

**Strengths:**

1.The paper identifies a significant issue with current contrastive self-supervised learning methods in their indiscriminate treatment of semantic-altering and semantics-preserving augmentations, which can lead to performance degradation in downstream tasks.
2. The paper provides a detailed theoretical explanation and empirical evidence for the importance of distinguishing between these two types of augmentations.
3.The paper introduces Direct Intra-image Contrastive Regularization (DICR), a method that aims to ensure the distinguishability of different but co-occurring contents through intra-image contrast, while preserving their correlation through inter-image contrast.
4. The paper demonstrates DICR's ability to differentiate between various semantic contents within an image and its effectiveness in enhancing the generalizable performance of existing methods in downstream tasks.

**Weaknesses:**

1.The paper assumes that distinguishing between semantic-altering and semantics-preserving augmentations will enhance generalizability, but more experiments may be needed to validate this assumption across different datasets and tasks.
2.Although DICR does not require additional parameters, it may increase the computational cost of training, especially with large datasets.
3.The introduction of DICR may complicate the model training process, requiring careful tuning to ensure performance.

**Questions:**

1.The effectiveness of DICR may rely heavily on the choice of augmentations, and different choices could affect model performance.
2.DICR's emphasis on distinguishing different contents could lead to overfitting to specific types of datasets, potentially underperforming on unseen distributions.

---

### Official Review · Reviewer_26Bi · 2024-11-04

**Soundness:** 1
**Presentation:** 3
**Contribution:** 2
**Rating:** 3
**Confidence:** 4

**Summary:**

This paper studies image-based contrastive learning. Observing that existing contrastive learning methods suffer from semantic change in applying croppings, the paper proposes DICR, a PnP regularization method that is explicitly designed to solve this issue. The experiments show that the proposed method has improved in various settings and downstream tasks.

**Strengths:**

- The proposed method is novel, simple yet effective. By only adding a branch of loss after the feature maps, the proposed method can mitigate the issue.
- The method is described in a clear manner with lots of formulas, making it easy to understand.
- The proposed method consistently outperforms the w/o baselines in most evaluation cases. Ablation studies are also provided.

**Weaknesses:**

- Lack lots of empirical comparisons to extensively verify the effectiveness of the method.
   - All the pre-training datasets are very small datasets. It is unsure whether the proposed method can work on large-scale datasets like ImageNet.
   - The only backbone is ResNet18. It is also uncertain whether the proposed method can also work on (1) larger ResNets, or (2) other model architectures like ViTs.
   - The only downstream task is classification. It is uncertain whether the method can improve the performance in more semantic-aware tasks like segmentation or captioning.
   - Not all common contrastive learning methods are considered, e.g., MoCo v2 and MoCo v3. Especially MoCo v3, which got rid of the queues and achieved contrastive learning with large batch sizes.
   - This lack of comparison significantly hurts the soundness of the paper. If the proposed method only works on small datasets with small networks, the conclusion is not very reasonable and cannot be applied on most of the scenarios, and the contribution will be limited.
- Some abbreviations are not clearly described. The audience needs to make educated guesses to find out what it means. E.g., "RoI" - "region of interests?"
- L083 claims that "which adds no extra parameters and can efficiently regularizes existing methods." However, no information about "efficiency" was provided as experiment results. I wonder what the actual additional computation cost is.

**Questions:**

- Please refer to the weakness to see some additional experiments that can be added to improve the soundnesses.
- I wonder whether other non-contrastive self-supervised learning, e.g., MAE or DINO, can also benefit from the proposed method.

---

### Note · Authors · 2024-11-26

I have read and agree with the venue's withdrawal policy on behalf of myself and my co-authors.